# SINKHORN AUTOENCODERS

## ABSTRACT

Optimal Transport offers an alternative to maximum likelihood for learning generative autoencoding models. We show how this principle dictates the minimization of the Wasserstein distance between the encoder aggregated posterior and the prior, plus a reconstruction error. We prove that in the non-parametric limit the autoencoder generates the data distribution if and only if the two distributions match exactly, and that the optimum can be obtained by deterministic autoencoders. We then introduce the Sinkhorn AutoEncoder (SAE), which casts the problem into Optimal Transport on the latent space. The resulting Wasserstein distance is minimized by backpropagating through the Sinkhorn algorithm. SAE models the aggregated posterior as an implicit distribution and therefore does not need a reparameterization trick for gradients estimation. Moreover, it requires virtually no adaptation to different prior distributions. We demonstrate its flexibility by considering models with hyperspherical and Dirichlet priors, as well as a simple case of probabilistic programming. SAE matches or outperforms other autoencoding models in visual quality and FID scores.

## 1 INTRODUCTION

Unsupervised learning aims to find the underlying rules that govern a given data distribution. It can be approached by learning to mimic the data generation process, or by finding an adequate representation of the data. Generative Adversarial Networks (GAN) (Goodfellow et al., 2014) belong to the former class, by learning to transform noise into a distribution that matches the given one. AutoEncoders (AE) (Hinton & Salakhutdinov, 2006) are of the latter type, by learning a representation that maximizes the mutual information between the data and its reconstruction, subject to an information bottleneck. Variational AutoEncoders (VAE) (Kingma & Welling, 2013; Rezende et al., 2014), provide both a generative model — i.e. a *prior* distribution on the latent space with a decoder that models the conditional likelihood — and an encoder — approximating the *posterior* distribution of the generative model. Optimizing the exact marginal likelihood is intractable in latent variable models such as VAE's. Instead one maximizes the Evidence Lower BOund (ELBO) as a surrogate. This objective trades off a reconstruction error of the input and a regularization term that aims at minimizing the Kullback-Leibler (KL) divergence from the approximate posterior to the prior.

An alternative principle for learning generative autoencoders is proposed by Tolstikhin et al. (2018). The theory of Optimal Transport (OT) (Villani, 2008) prescribes a different regularizer: one that matches the prior with the *aggregated posterior* — the average (approximate) posterior over the training data. In Wasserstein AutoEncoders (WAE) (Tolstikhin et al., 2018), this is enforced by the choice of either the Maximum Mean Discrepancy (MMD) (Gretton et al., 2012)), or by adversarial training on the latent space. WAE empirically improves upon VAE. More recently, a family of Wasserstein divergences has been used by Ambrogioni et al. (2018) in the context of variational inference. The particular choice of Wasserstein distances may be crucial for convergence, due to the induced weaker topology as compared to other divergences, such as the KL (Arjovsky et al., 2017).

We contribute to the formal analysis of autoencoders with OT. First, we prove that in order to minimize the Wasserstein distance between the generative model and the data distribution, we can minimize the usual reconstruction-plus-regularizer cost, where the regularizer is the Wasserstein distance between the encoder aggregated posterior and the prior. Second, in the non-parametric limit, the model learns the data distribution *if and only if* the aggregated posterior matches the prior exactly. Third, as a consequence of the Monge-Kontorovich equivalence (Villani, 2008), the functional space of this learning problem can be limited to that of deterministic autoencoders.

The theory supports practical innovations. We learn deterministic autoencoders by minimizing a reconstruction error and the Wasserstein distance on the latent space between samples of the aggregated posterior and the prior. The latter is known to be costly, but a fast approximate solution is provided by the Sinkhorn algorithm (Cuturi, 2013). We follow Frogner et al. (2015) and Genevay et al. (2018), by exploiting the differentiability of the Sinkhorn iterations, and unroll it for backpropagation. Altogether, we call our method the Sinkhorn AutoEncoder (SAE).

The Sinkhorn AutoEncoder is agnostic to the analytical form of the prior, as it optimizes a sample-based cost function which is aware of the geometry of the latent space. Furthermore, as a byproduct of using deterministic networks, it models the aggregated posterior as an implicit distribution (Mohamed & Lakshminarayanan, 2016) with no need of the reparametrization trick for learning the encoder (Kingma & Welling, 2013). Therefore, with essentially no change in the algorithm, we can learn models with Normally distributed priors and aggregated posteriors, as well as distributions living on manifolds such as hyperspheres (Davidson et al., 2018) and probability simplices.

We start our experiments by studying unsupervised representation learning by training an encoder in isolation. Our results demonstrate the capability of the Sinkhorn algorithm to produce embeddings that conserve the local geometry of the data, echoing results from Bojanowski & Joulin (2017). Next we move to the autoencoder. In an ablation study, we compare with the exact Hungarian algorithm in place of the Sinkhorn and show that our method performs equally well, while converging faster. We then compare against prior work on autoencoders with Normal and spherical priors on MNIST, CIFAR10 and CelebA. SAE with a spherical prior produces visually more appealing interpolations, crisper samples and comparable or lower FID (Heusel et al., 2017). Finally, we further show the flexibility of SAE with qualitative results by using a Dirichlet prior, which defines the latent space on a probability simplex, as well as with a simple probabilistic programming task.

## 2 BACKGROUND

### 2.1 WASSERSTEIN DISTANCE AND WASSERSTEIN AUTOENCODERS

We follow Tolstikhin et al. (2018) and denote with $\mathcal{X}, \mathcal{Y}, \mathcal{Z}$ the sample spaces and with $X, Y, Z$ and $P_X, P_Y, P_Z$ the corresponding random variables and distributions. Given a map $F : \mathcal{X} \to \mathcal{Y}$ we denote by $F_\#$ the push-forward map acting on a distribution $P$ as $P \circ F^{-1}$. If $F(Y|X)$ is non-deterministic we define the push-forward of a distribution $P$ as the induced marginal of the joint distribution $F(Y|X)P_X$ (denoted by $F(Y|X)_\# P_X$). For any measurable non-negative *cost* $c : \mathcal{X} \times \mathcal{Y} \to \mathbb{R}^+ \cup \{\infty\}$, one can define the following *OT-cost* between marginal distributions $P_X$ and $P_Y$ via:

$$W_c(P_X, P_Y) = \inf_{\Gamma \in \Pi(P_X, P_Y)} \mathbb{E}_{(X,Y) \sim \Gamma}[c(X, Y)], \tag{1}$$

where $\Pi(P_X, P_Y)$ is the set of all joint distributions that have as marginals the given $P_X$ and $P_Y$. The elements from $\Pi(P_X, P_Y)$ are called *couplings* from $P_X$ to $P_Y$. From now on we will assume that $\mathcal{X} = \mathcal{Y}$ and $c(x, y)$ is a distance. In this case $W_c(P_X, P_Y)$ is the Wasserstein distance w.r.t the cost $c$. If $c(x, y) = \|x - y\|_p^p$ for $p \geq 1$ then $W_p = \sqrt[p]{W_c}$ is called the *p-th Wasserstein distance*.

Let $P_X$ denote the true data distribution on $\mathcal{X}$. We define a *latent variable model* given as follows: we fix a latent space $\mathcal{Z}$ and a prior distribution $P_Z$ on $\mathcal{Z}$ and consider the conditional distribution $G(X|Z)$ (the decoder) parameterized by a neural network $G$. Together they specify a generative model as $G(X|Z)P_Z$. The induced marginal will be denoted by $P_G$. Learning $P_G$ to approximate the true $P_X$ is then defined as:

$$\min_G W_c(P_X, P_G).$$

Because of the infimum over $\Pi(P_X, P_G)$ inside $W_c$, this is intractable. To rewrite this objective we consider the posterior distribution $Q(Z|X)$ (the encoder) and its *aggregated posterior* $Q_Z$:

$$Q_Z = Q(Z|X)_\# P_X = \mathbb{E}_{X \sim P_X} Q(Z|X), \tag{2}$$

the induced marginal of the joint distribution $Q(Z|X)P_X$. Tolstikhin et al. (2018) show that, if the decoder $G(X|Z)$ is deterministic, i.e. $P_G = G_\# P_Z$, or in other words, if all stochasticity of the generative model is captured by $Z$, then:

$$W_c(P_X, P_G) = \inf_{Q(Z|X): \ Q_Z = P_Z} \mathbb{E}_{X \sim P_X} \mathbb{E}_{Z \sim Q(Z|X)}[c(X, G(Z))]. \tag{3}$$

Learning the generative model $G$ with the Wasserstein AutoEncoder amounts to:

$$\min_G \min_{Q(Z|X)} \mathbb{E}_{X \sim P_X} \mathbb{E}_{Z \sim Q(Z|X)}[c(X, G(Z))] + \beta \cdot D_Z(Q_Z, P_Z), \tag{4}$$

where $\beta > 0$ is a Lagrange multiplier and $D_Z$ is any divergence measure on probability distributions on $\mathcal{Z}$, which choice is left open. WAE uses either MMD or a discriminator trained adversarially for $D_Z$. As discussed in Bousquet et al. (2017), Equation 4 is a lower bound of Equation 3 for any value of $\beta > 0$. Minimizing this lower bound does not ensure a minimization of the original objective of Equation 3.

## 2.2 THE SINKHORN ALGORITHM

In place of any choice of $D_Z$, in Section 3 we formally support the minimization of a *Wasserstein distance on latent space*. The distance is notoriously hard to compute, which is the reason why the rewriting of Equation 3 is of practical interest. When restricting to discrete distributions, the problem becomes more amenable and efficient approximations exist. To motivate this direction, recall that we can always see *samples* of a continuous distribution as Dirac deltas, whose expectation defines a discrete distribution. Let two discrete distributions with support on $M$ points be $\hat{P} = \frac{1}{M} \sum_{i=1}^{M} \delta_{z_i}$, $\hat{Q} = \frac{1}{M} \sum_{i=1}^{M} \delta_{z'_i}$. Given a cost $c'$, their (empirical) Wasserstein distance is:

$$W_{c'}(\hat{Q}, \hat{P}) = \min_{R \in S_M} \frac{1}{M} \langle R, C' \rangle_F, \tag{5}$$

where $C'_{ij} = c'(z'_i, z_j)$ is the matrix associated to the cost $c'$, $R$ is a doubly stochastic matrix as defined in $S_M = \{R \in \mathbb{R}_{\geq 0}^{M \times M} \mid R\mathbf{1} = \mathbf{1}, R^T\mathbf{1} = \mathbf{1}\}$, and $\langle \cdot, \cdot \rangle_F$ denotes the Frobenius inner product; $\mathbf{1}$ is the vector of ones. Eq. (5) is known to converge to the Wasserstein distance between the continuous distributions as $M$ tends to infinity (Weed & Bach, 2017). This linear program has solutions on the vertices of $S_M$, which is the set of permutation matrices (Peyré & Cuturi, 2018). The Hungarian algorithm finds an optimal solution in $O(M^3)$ time (Kuhn, 1955).

An entropy-regularized version of problem (5) can be solved more efficiently. Let the entropy of $R$ be $H(R) = -\sum_{i,j=1}^{M} R_{i,j} \log R_{i,j}$. For $\varepsilon > 0$, Cuturi (2013) defines the *Sinkhorn distance $S_{c'}$*:

$$R^* = \arg\min_{R \in S_M} \frac{1}{M} \langle R, C' \rangle_F - \varepsilon H(R), \qquad S_{c'}(\hat{Q}, \hat{P}) = \langle R^*, C' \rangle_F, \tag{6}$$

and shows that the (Sinkhorn, 1964)'s algorithm returns its regularized optimum — that is also unique due to strong convexity of the entropy. The Sinkhorn is a fixed point algorithm that runs nearly in $M^2$ time (Altschuler et al., 2017) and can be efficiently implemented with matrix multiplications; see Algorithm 1. Its convergence to the Wasserstein distance is studied by Weed (2018).

The smaller the $\varepsilon$, the smaller the entropy and the better the approximation of the Wasserstein distance. At the same time, a larger number of steps $O(L)$ is needed to converge. Conversely, high entropy encourages the solution to lie far from a permutation matrix. Note that all Sinkhorn operations are differentiable. So when the distance is used as a cost function, we can unroll $O(L)$ iterations and backpropagate (Genevay et al., 2018). In conclusion, we obtain a differentiable surrogate for Wasserstein distances between empirical distributions; the approximation arises from sampling, entropy regularization and the finite amount of steps in place of convergence.

## 2.3 NOISE AS TARGETS

Bojanowski & Joulin (2017) introduce Noise As Targets (NAT), an algorithm for unsupervised representation learning. The method learns a neural network $f_\theta$ by embedding images into a uniform hypersphere. A sample $z$ is drawn from the sphere for each training image and fixed. The goal is to learn $\theta$ such that 1-to-1 matching between images and samples is improved: matching is coded with a permutation matrix $R$, and updated with the Hungarian algorithm. The objective is:

$$\max_\theta \max_{R \in P_M} \text{Tr}(R Z f_\theta(X)^\top), \tag{7}$$

where $\text{Tr}(\cdot)$ is the trace operator, $Z$ and $X$ are respectively prior samples and images stacked in a matrix and $P_M \subset S_M$ is the set of $M$-dimensional permutations. NAT learns by alternating SGD and the Hungarian. One can interpret this problem as supervised learning, where the samples are targets (sampled only once) but their assignment is learned; notice that freely learnable $Z$ would make the problem ill-defined. The authors relate NAT to OT, a link that we make formal below.

## 3 PRINCIPLES OF WASSERSTEIN AUTOENCODING

With Equation 3, Tolstikhin et al. (2018) reformulate the Wasserstein distance in image space in terms of autoencoders. The hard constraint $Q_Z = P_Z$ is in practice replaced with a soft constraint by adding a penalty in the form of a divergence $D_Z(Q_Z, P_Z)$. The resulting objective (4) is a lower bound to the Wasserstein difference and the choice of a divergence $D_Z$ is left open. In contrast, we show that one should opt for minimizing a Wasserstein distance in latent space and that this leads to an equality with — not a bound for — the original Wasserstein distance in image space.

More precisely, Theorem 3.1 first proves that the Wasserstein distance between the generative model and data distribution is bounded from above by a quantity consisting of the reconstruction error and the Wasserstein distance between $P_Z$ and $Q_Z$. Theorem 3.2 shows that we can restrict learning to the class of *deterministic* (auto)encoders. Put together, Corollary 3.3 provides a principled learning objective in the framework of Optimal Transport by rewriting the Wasserstein distance in image space into an equivalent tractable form. We start with the following bound:

**Theorem 3.1.** *If $G(X|Z)$ is deterministic and $\gamma$-Lipschitz then:*

$$W_p(P_X, P_G) \leq W_p(P_X, G_\# Q_Z) + \gamma \cdot W_p(Q_Z, P_Z).$$

The proof (A.2) exploits the triangle inequality of the Wasserstein distance and its behaviour under composition with Lipschitz maps – a property not shared with divergences such as the KL. To effectively minimize the right-hand side in Theorem 3.1 over a class of encoders we need to further upper bound the reconstruction term with the following[1]:

$$W_p(P_X, G_\# Q_Z) \leq \sqrt[p]{\mathbb{E}_{X \sim P_X} \mathbb{E}_{Z \sim Q(Z|X)} \mathbb{E}_{X' \sim G(X|Z)}[\|X - X'\|_p^p]}, \tag{8}$$

which reduces to the $p$-th root of $\mathbb{E}_{X \sim P_X}[\|X - G(Q(X))\|_p^p]$ if both $G$ and $Q$ are deterministic. The tightness of this bound and its use as an objective function for learning are discussed below.

We now improve the characterization of Equation 3, which is formulated in terms of stochastic encoders $Q(Z|X)$ and deterministic decoders $G(X|Z)$. In fact, it is possible to restrict the learning class to that of deterministic *autoencoders*:

**Theorem 3.2.** *Let $P_X$ be not atomic[2] and $G(X|Z)$ deterministic. Then for every continuous cost $c$:*

$$W_c(P_X, P_G) = \inf_{Q(Z|X) \ deterministic:\ Q_Z = P_Z} \mathbb{E}_{X \sim P_X}[c(X, G(Q(X)))].$$

*Using the cost $c(x, y) = \|x - y\|_p^p$, the equation holds with $W_p^p(P_X, P_G)$ in place of $W_c(P_X, P_G)$.*

The statement is a direct consequence of the equivalence between the Kantorovich and Monge formulations of OT (Villani, 2008); see the proof in A.3. We remark that this result is stronger than, and can be used to deduce Equation 3; see A.4 for a proof. Combining the two previous results, we are now in position to prove that the bound in Theorem 3.1 is tight for deterministic (auto)encoders:

$$W_p(P_X, P_G) \overset{Th.3.1}{\leq} \inf_{Q \ det.} \sqrt[p]{\mathbb{E}_{X \sim P_X}[\|X - G(Q(X))\|_p^p]} + \gamma \cdot W_p(Q_Z, P_Z) \tag{9}$$

$$\leq \inf_{Q \ det., Q_Z = P_Z} \sqrt[p]{\mathbb{E}_{X \sim P_X}[\|X - G(Q(X))\|_p^p]} + \gamma \cdot \underbrace{W_p(Q_Z, P_Z)}_{=0} \tag{10}$$

$$\overset{Th.3.2}{=} W_p(P_X, P_G). \tag{11}$$

Inequality in Step 10 holds because we restrict the domain of the *infimum*, which in turns implies $W_p(Q_Z, P_Z) = 0$. As a consequence we obtain the following Corollary, which provides us with an objective for learning generative autoencoders:

---

[1] This is from the fact that $(\mathrm{id}_\mathcal{X}, G)_\# Q(Z|X)P_X \in \Pi(P_X, G_\# Q_Z)$.

[2] A probability measure is non-atomic if every point in its support has zero measure. It is important to distinguish between the *empirical* data distribution $\hat{P}_X$, which is always atomic, and the underlying *true* distribution $P_X$, only which we need to assume to be non-atomic.

**Corollary 3.3.** *Let $P_X$ be non-atomic and $G(X|Z)$ be deterministic and $\gamma$-Lipschitz. Then we have the equality:*

$$W_p(P_X, P_G) = \inf_{Q(Z|X) \; deterministic} \sqrt[p]{\mathbb{E}_{X \sim P_X}[\|X - G(Q(X))\|_p^p]} + \gamma \cdot W_p(Q_Z, P_Z). \quad (12)$$

More precisely, we can now formulate our learning problem as the minimization of the right-hand side of Equation 12 over deterministic decoders:

$$\min_G \min_Q \sqrt[p]{\mathbb{E}_{X \sim P_X}[\|X - G(Q(X))\|_p^p]} + \gamma \cdot W_p(Q_Z, P_Z) \quad (13)$$

When the encoder is a neural network of limited capacity, enforcing $Q_Z \approx P_Z$ might not be feasible in the general case of dimension mismatch (Rubenstein et al., 2018). In fact, since the class of deterministic neural networks is much smaller than the class of deterministic measurable maps, one might consider adding noise to the output, i.e. use stochastic networks instead. Nonetheless, neural networks can approximate any measurable map up to arbitrarily small error (Hornik, 1991), and we prove a related bound for the Wasserstein distance in A.5. It follows that learning deterministic autoencoders is sufficient to approach the theoretical upper bound and thus it will be our empirical choice.

Finally, Theorems 3.1 and 3.2 strengthen the relevance of matching aggregated posterior and prior, which we show to be a sufficient and necessary condition for generative autoencoding. Justified by the previous results, we state it for deterministic autoencoders (proof in A.6).

**Theorem 3.4** (Sufficiency and necessity for generative autoencoding). *Suppose perfect reconstruction, that is, $P_X = (G \circ Q)_\# P_X$. Then:*

$$i) \; P_Z = Q_Z \implies P_X = P_G, \qquad ii) \; P_Z \neq Q_Z \implies P_X \neq P_G. \quad (14)$$

In particular, Theorem 3.4 ii) certifies that, under perfect reconstruction, *failing* to match aggregated posterior and prior makes learning the data distribution impossible. Matching in latent space should be seen as fundamental as minimizing the reconstruction error, a fact known about the performance of VAE (Hoffman & Johnson, 2016; Higgins et al., 2017; Alemi et al., 2018; Rosca et al., 2018).

## 4    SINKHORN AUTOENCODERS

In light of our theory, we minimize the Wasserstein distance between the aggregated posterior and the prior, and we do so by running the Sinkhorn on their empirical samples. Let $\{x_i\}_{i=1}^M$ be the data input to the deterministic encoder $Q(z_i'|x_i) = \delta_{z_i'}$ and $\{z_i\}_{i=1}^M$ the samples from the prior $P_Z$. The empirical distributions are $\hat{Q}_Z = \frac{1}{M} \sum_{i=1}^M \delta_{z_i'}$ and $\hat{P}_Z = \frac{1}{M} \sum_{i=1}^M \delta_{z_i}$. With $C_{ij}' = c(z_i', z_j)$, the Sinkhorn distance is $S_{c'}(\hat{Q}_Z, \hat{P}_Z)$ as defined in Equation 6.

We compute the Sinkhorn distance in two steps: first obtain the optimal regularized coupling $R^*$ and then multiply it with the cost, i.e. set $\varepsilon = 0$:

$$R^* = \arg\min_{R \in \mathbb{S}_M} \frac{1}{M} \langle R, C' \rangle_F - \varepsilon H(R)$$

$$S_{c'}(\hat{Q}_Z, \hat{P}_Z) = \frac{1}{M} \langle R^*, C' \rangle_F . \quad (15)$$

See Algorithm 1. Note that we do not sacrifice differentiability: we stack $O(L)$ Sinkhorn operations on top of the encoder, without additional learnable parameters, and run auto-differentiation.

---
**Algorithm 1** SINKHORN

**Input:** $\{z_i\}_{i=1}^m \sim P_Z$, $\{z_i'\}_{i=1}^m \sim Q_Z$, $\varepsilon, L$
$\forall i, j, \; C_{ij} = c(z_i, z_j')$
$K = e^{-C/\varepsilon}, u \leftarrow \mathbf{1}$          # elem-wise exp
**repeat** $L$ **times**:
   $v \leftarrow \mathbf{1}/(K^\top u)$          # elem-wise division
   $u \leftarrow \mathbf{1}/(Kv)$
$R^* \leftarrow \text{Diag}(u)K\text{Diag}(v)$
**Output:** $\frac{1}{M} \langle R^*, C \rangle_F$

---

With a deterministic decoder $G$ and encoder $Q$, we arrive at the objective for the Sinkhorn AutoEncoder (SAE):

$$\min_G \min_Q \mathbb{E}_{X \sim \hat{P}_X}[\|X - G(Q(X))\|_p^p] + \beta \cdot S_{c'}(\hat{Q}_Z, \hat{P}_Z). \quad (16)$$

In practice, we drop the $p$-th root and tune a $\beta > 0$ hyper-parameter to mix the two terms. Small $\varepsilon$ and hence large $L$ worsen the numerical stability of the Sinkhorn; thus it is more convenient to scale $S_{c'}$ by $\beta$ to explore the trade off, as in the WAE. In most experiments, both $c$ and $c'$ will be $\| \cdot \|_2^2$. This objective is minimized by mini-batch SGD, which requires the re-calculation of an optimal regularized coupling $R^*$ at each iteration. Experimentally we found that this is not a significant overhead, unless a large $L$ is needed for convergence due to a small $\varepsilon$. In practice, Algorithm 1 loops for $L$ iterations but can exit earlier if the updates of $u$ reach a fixed point.

We have not specified our distribution $P_Z$ yet. In fact, SAE can work in principle with arbitrary priors. The only requirement coming from the Sinkhorn is the ability to generate samples. The choice should be motivated by the desired geometric properties of the latent space; Theorem 3.4 stresses the importance of such choice for the generative model. For quantitative comparison with prior work, we focus primarily on hyperspheres, as in the Hyperspherical VAE (HVAE) (Davidson et al., 2018). Moreover, considering the Wasserstein distance ($\varepsilon = 0$) from a uniform hyperspherical prior with squared Euclidean cost, we recover the NAT objective as a special case of ours (see Appendix A.7); yet, our method enjoys lower complexity and differentiability. The remarkable performance of NAT on representation learning on ImageNet confirms the value of the spherical prior. Other distributions are also considered in the paper, in particular the Dirichlet prior — with a tunable bias towards the simplex vertices — as a choice for controlling latent space clustering.

Deterministic encoders model implicit distributions. Distributions are said to be implicit when their probability density may be intractable or even unknown, but it is possible to obtain samples and gradients for their parameters; GANs are examples of models with implicit distributions. Implicit distributions can give more flexibility as they are not limited by families of distributions with tractable density (Mohamed & Lakshminarayanan, 2016; Huszár, 2017). Moreover, by encoding with deterministic neural networks, we bypass the use of reparametrization tricks for gradient estimation.

## 5 RELATED WORK

The normal prior is common in VAE for the reason of tractability. In fact, changing the prior and/or the approximate posterior distributions requires the use of tractable densities and the appropriate reparametrization trick. A hyperspherical prior is used by Davidson et al. (2018) with improved experimental performance; the algorithm models a Von Mises-Fisher posterior, with a non-trivial posterior sampling procedure and a reparametrization trick based on rejection sampling. Our implicit encoder distribution sidesteps these difficulties; recent advances on variable reparametrization can also simplify these requirements (Figurnov et al., 2018). We are not aware of methods embedding on probability simplices, except the use of Dirichlet priors by the same Figurnov et al. (2018).

Hoffman & Johnson (2016) showed that the objective of a VAE does not force the aggregated posterior and prior to match, and that the mutual information of input and codes may be minimized instead. Just like the WAE, SAE avoids this effect by construction. Makhzani et al. (2015) and WAE improve latent matching by GAN/MMD. With the same goal, Alemi et al. (2017), Tomczak & Welling (2017) introduce learnable priors in the form of a mixture of approximate posteriors, which can be used in SAE as well.

The Sinkhorn (1964) algorithm gained interest after Cuturi (2013) showed its application for fast computation of Wasserstein distances. The algorithm has been applied to ranking (Adams & Zemel, 2011), domain adaptation (Courty et al., 2014), multi-label classification (Frogner et al., 2015), metric learning (Huang et al., 2016) and ecological inference (Muzellec et al., 2017). Santa Cruz et al. (2017); Linderman et al. (2018) used it for supervised combinatorial losses. Our use of the Sinkhorn for generative modeling is akin to that of Genevay et al. (2018), which matches data and model samples with adversarial training, and to Ambrogioni et al. (2018), which matches samples from the model joint distribution and a variational joint approximation. WAE and WGAN objectives are linked respectively to primal and dual formulations of OT (Tolstikhin et al., 2018).

Our approach for training the encoder alone qualifies as self-supervised representation learning (Donahue et al., 2017; Noroozi & Favaro, 2016; Noroozi et al., 2017). As in NAT (Bojanowski & Joulin, 2017) and in constrast to most other methods, we can sample pseudo labels (from the prior) independently from the input. In Appendix A.7 we show a formal connection with NAT.

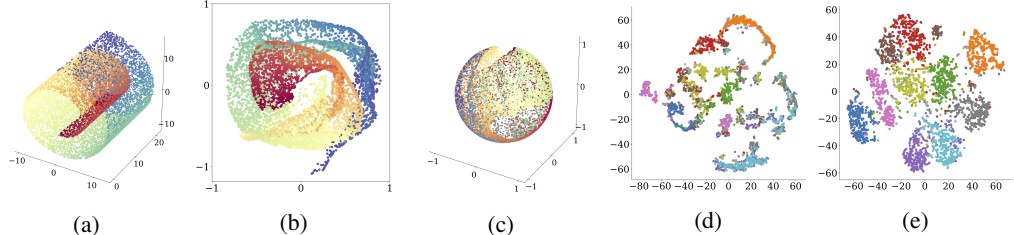

Figure 1: a) Swiss Roll and its b) squared and c) spherical embeddings learned by Sinkhorn encoders. MNIST embedded onto a 10D sphere viewed through $t$-SNE, with classes by colours: d) encoder only or e) encoder + decoder.

## 6 EXPERIMENTS

We start our empirical analysis with a qualitative assessment of the representation learned with the Sinkhorn algorithm. In the rest we focus on the autoencoder. We compare with NAT and confirm the Sinkhorn to be a better choice than the Hungarian. We display interpolations and samples of SAE and compare numerically with AE, ($\beta$)-VAE, HVAE and WAE-MMD. We further show the flexibility of SAE by using a Dirichlet prior and on a toy probabilistic programming task.

We experiment on MNIST, CIFAR10 (Krizhevsky & Hinton, 2009) and CelebA (Liu et al., 2015). MNIST is dynamically binarized and the reconstruction error is the binary cross-entropy (although not a distance, it is a commonly used divergence for binary data). For CIFAR10 and CelebA the reconstruction is the squared Euclidean distance; in every experiment, the latent cost is also squared Euclidean. We train fully connected neural networks for MNIST and the convolutional architectures from Tolstikhin et al. (2018) for the rest; the latent space dimensions are respectively 10, 64, 64. We run Adam (Kingma & Ba, 2014) with mini-batches of 128. Hyperspherical embedding is hardcoded in the architectures by $L2$ normalization of the encoder output as in Bojanowski & Joulin (2017). The Sinkhorn runs with $\epsilon = 0.1$, $L = 50$, except when otherwise stated. FID scores for CIFAR10 and CelebA are calculated as in Heusel et al. (2017), while for MNIST we train a 2-layer convolutional network to extract features for the Fréchet distance, similarly to Odena et al. (2018). Notice that the FID score *is* a Wasserstein-2 distance and hence our theory applies directly.

### 6.1 REPRESENTATION LEARNING WITH SINKHORN ENCODERS

We demonstrate qualitatively that the Sinkhorn distance is a valid objective for unsupervised feature learning by showing that we can learn the encoder in isolation. The task is to embed the input distribution in a lower dimensional space, preserving the local data geometry, by solving Problem 14 with no reconstruction cost. We display the representation of a 3D Swiss Roll and MNIST. For the Swiss Roll we set $\varepsilon = 10^{-3}$, while for MNIST it is set to 0.5, and $L$ is picked to ensure convergence. For the Swiss roll (Figure 1a), we use a 50-50 fully connected network with ReLUs.

Figures 1b, 1c show that the local geometry of the Swiss Roll is conserved in the new representational spaces — a square and a sphere. While the global shape is not necessarily more unfolded than the original, it looks qualitatively more amenable for further computation. Figure 1d shows the $t$-SNE visualization (Maaten & Hinton, 2008) of the learned representation of the MNIST test set. With neither labels nor reconstruction error, we learn an embedding that is aware of class-wise clusters. Minimization of the Sinkhorn distance achieves this by encoding onto a $d$-dimensional uniform sphere, such that points are encouraged to map far apart; in particular, in high dimension we can prove (see A.8) that the collapse probability decreases with $d$:

**Proposition 6.1.** *Let $z, z'$ be two uniform samples from a $d$-dimensional sphere. In the high dimensional regime, for any $\delta < \sqrt{2}$ we have $P(\|z - z'\|_2 > \delta) \geq 1 - \frac{1}{4d(\sqrt{2}-\delta)^2}$.*

Other than this repulsive effect — the uniform distribution has max-entropy on any compact space —, a contractive force is present due to the inductive prior of neural networks, which are known to be Lipschitz functions (Balan et al., 2017). On the one hand, points in the latent space disperse in order to fill up the sphere; on the other hand, points close on image space cannot be mapped too far from

| | | | MNIST | | | | CIFAR10 | | |
|---|---|---|---|---|---|---|---|---|---|
| method | prior | $\beta$ | MMD | RE | FID | $\beta$ | MMD | RE | FID |
| Hungarian | sample | 10 | 0.37 | 65.9 | 10.3 | 10 | 0.25 | 22.4 | 98.5 |
| Hungarian | targets | 10 | 0.32 | 68.5 | 10.0 | 10 | 0.26 | 22.8 | 98.4 |
| Hungarian | sample | 100 | 0.60 | 85.0 | 9.7 | 100 | 0.23 | 23.8 | 98.6 |
| Hungarian | targets | 100 | 0.21 | 67.2 | 7.1 | 100 | 0.24 | 23.5 | 102.0 |
| Sinkhorn | sample | 10 | 0.35 | 66.2 | 9.4 | 10 | 0.25 | 22.5 | 97.5 |
| Sinkhorn | targets | 10 | 0.29 | 65.3 | 9.4 | 10 | 0.25 | 22.4 | 97.0 |
| Sinkhorn | sample | 100 | 0.30 | 66.8 | 6.8 | 100 | 0.21 | 23.7 | 100.4 |
| Sinkhorn | targets | 100 | 0.30 | 66.8 | 6.8 | 100 | 0.24 | 23.1 | 107.5 |

Table 1: Ablation for spherical SAE: Sinkhorn vs. Hungarian, fixed targets vs. sampling. MMD are scaled up by 1000. We compute a baseline for the MMD between two independent set of 10K samples (same as the test set size) from the prior. The baseline is 0.2 for both datasets.

| | | | MNIST | | | | CIFAR10 | | | | CelebA | | |
|---|---|---|---|---|---|---|---|---|---|---|---|---|---|
| method | prior | cost | $\beta$ | MMD | RE | FID | $\beta$ | MMD | RE | FID | $\beta$ | MMD | RE | FID |
| AE | - | - | - | - | 62.6 | 45.2 | - | - | 22.6 | 375.0 | - | - | 61.8 | 357.0 |
| VAE | normal | KL | 1 | 0.63 | 66.4 | **7.2** | 1 | 4.6 | 40.6 | 161.0 | 1 | 0.35 | 75.1 | **51.4** |
| $\beta$-VAE | normal | KL | 0.1 | 2.3 | 62.8 | 15.2 | 0.1 | 0.23 | 22.8 | 106.6 | 0.1 | 0.21 | 63.7 | **56.5** |
| WAE | normal | MMD | 100 | 0.69 | 63.1 | 9.0 | 100 | 0.29 | 22.9 | 105.3 | 100 | 0.21 | 62.6 | 61.6 |
| AE | sphere[†] | - | - | 4.7 | 66.2 | 22.0 | - | 1.8 | 22.4 | 107.8 | - | 1.1 | 62.4 | 83.9 |
| HVAE | sphere | KL | 1 | 0.33 | 72.2 | 9.5 | - | - | - | - | - | - | - | - |
| WAE | sphere | MMD | 100 | 0.25 | 65.7 | 8.9 | 100 | 0.24 | 22.4 | **99.7** | 100 | 0.23 | 61.9 | 61.3 |
| SAE | sphere | Sinkhorn | 100 | 0.30 | 66.8 | **6.8** | 10 | 0.23 | 22.5 | **97.2** | 10 | 0.26 | 63.4 | **56.5** |

Table 2: SAE vs. prior work. In boldface the best two FID per dataset. Note that MMD are not comparable if the prior is different. [†]The 'spherical' AE amounts to normalizing the encoder output.

each other. As a result, local distances are conserved while the overall distribution is spread. When the encoder is combined with a decoder $G$ — the topic of the experiments below —, the contractive force strenghtens: they collaborate in learning a latent space which makes reconstruction possible despite finite capacity and hence favours the conservation of local similarities; see Figure 1e.

## 6.2 AUTOENCODING WITH THE SINKHORN DISTANCE AND NAT

We investigate the advantages of the Sinkhorn with respect to NAT in training autoencoders; this is an ablation study for our method. First, Sinkhorn has a lower complexity than the Hungarian. In both cases, the complexity can be reduced by mini-batch optimization. Yet, training with large mini-batches ($> 200$) becomes quickly impractical with the Hungarian. Second, the differentiability of the Sinkhorn allows us to avoid the alternating minimization and instead backpropagate on the joint parameter space of encoder and doubly stochastic matrices. Third, the Sinkhorn approximates the empirical Wasserstein distance, while the Hungarian is optimal. Last, NAT draws samples once and uses them as targets throughout learning; their assignment to training points is updated by optimizing a permutation matrix over mini-batches and storing the local optimal result. We term NAT in this context *Hungarian-targets* and our method *Sinkhorn-sample*. We can design two hybrid methods. *Hungarian-sample*: a permutation $R$ can be used to compute the cost $\langle R, C' \rangle_F$ and backpropagate. *Sinkhorn-targets*: a doubly stochastic matrix $R$ solution of the Sinkhorn can be used for sampling a permutation[3] and targets can be re-assigned. We test the impact of these choices experimentally by test set reconstruction error and FID score on MNIST and CIFAR10; we measure latent space mismatch by the MMD with Gaussian kernel over the test set.

Table 1 shows the results. From the FID scores, we conclude that there is no significant difference in generative performance between either Sinkhorn vs. Hungarian, or samples vs. targets. The parameter $\beta$ trading off reconstruction and latent space cost is more influential than any of these choices. On MNIST, MMD is often lower with fixed targets; this is a sign that the FID does not fully account for all model qualities. Due to the additional overhead of the Hungarian and the targets updating, our algorithm implements the Sinkhorn with mini-batch sampling. In the rest, we also fix $\beta$ for MNIST and CIFAR as the best found here.

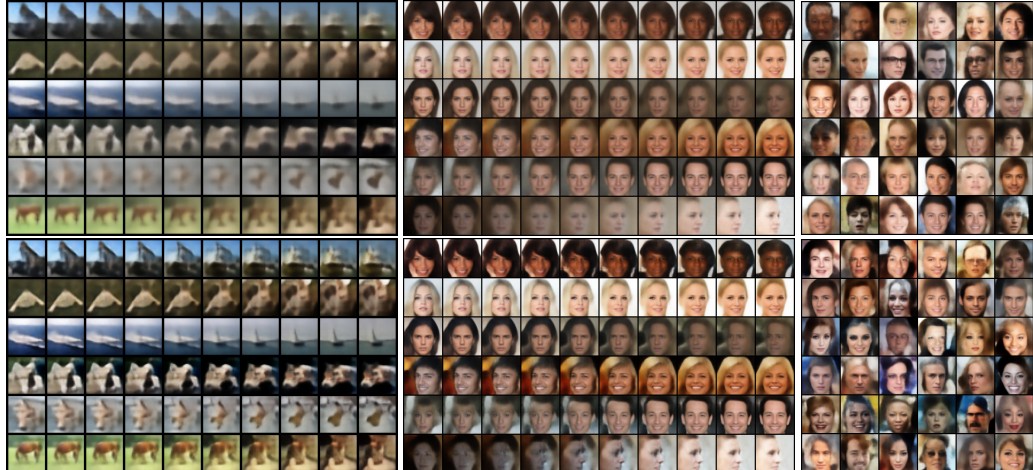

Figure 2: From left to right: CIFAR10 interpolations, CelebA interpolations and samples. Models from Table 2: ($\beta$-)VAE (top) and SAE (bottom).

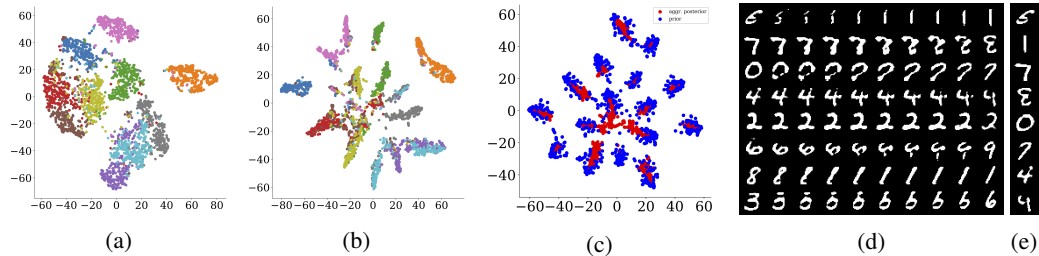

|     |     |     |     |     |
| --- | --- | --- | --- | --- |
| (a) | (b) | (c) | (d) | (e) |

Figure 3: $t$-SNEs of SAE latent spaces on MNIST: a) 10-dimensional $\mathrm{Dir}(1/2)$ and b) 16-dimensional $\mathrm{Dir}(1/5)$ priors. For the latter: c) aggr. posterior (red) vs. prior (blue), d) interpolation between vertices and e) samples from the prior.

### 6.3 COMPARISON WITH OTHER AUTOENCODERS

We compare with AE, ($\beta$-)VAE, HVAE[4] and WAE. Figures 2 shows interpolations and samples of SAE and VAE from CIFAR10 and CelebA. SAE interpolations are defined on geodesics connecting points on the hypersphere. SAE tends to produce crisper images, with higher contrast, and avoids averaging effects as particularly evident in the CelebA interpolations. The CelebA samples are also interesting: while SAE generally maintains a crisper look than VAE's, faces appear more often malformed. Table 2 reports a quantitative comparison. Each baseline model has a version with normal and spherical prior. FID scores of SAE are on par or superior to that of VAE and consistently better than WAE. The spherical prior appears to reduce FID scores in several cases.

### 6.4 DIRICHLET PRIORS

We further demonstrate the flexibility of SAE by using Dirichlet priors on MNIST. The prior draws samples on the probability simplex; hence, here we constrain the encoder by a final softmax layer. We use priors that concentrate on the vertices, by the intuition that digits would naturally cluster around them. A 10-dimensional $\mathrm{Dir}(1/2)$ prior (Figure 3a) results in an embedding qualitatively similar to the uniform sphere (1e). With a more skewed prior $\mathrm{Dir}(1/5)$, we could expect an organization in latent space where each digit is mapped to a vertex, as little mass lies in the center. We

---

[3]Obtaining the closest permutation to a double stochastic matrix is costly. We use a stochastic heuristic due to Fogel et al. (2013) that reduces to sorting. We select permutation minimizing $\langle C, \cdot \rangle_F$ out of 10 draws.

[4]Comparing with Davidson et al. (2018) in high dimension was unfeasible. The HVAE likelihood requires evaluating the Bessel function, which is computed on CPU. Note that SAE is oblivious to likelihood functions.

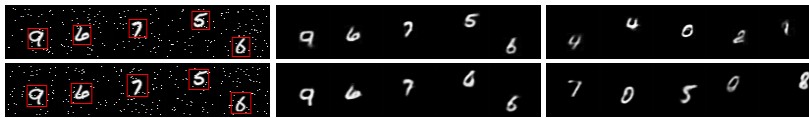

Figure 4: Toy probabilistic programming: data and localization (left), reconstructions (center) and samples (right). AIR (top) and SAE (bottom).

found that in dimension 10 this is seldom the case, as multiple vertices can be taken by the same digit to model different styles, while other digits share the same vertex.

We thus experiment with a 16-dimensional $\mathrm{Dir}(1/5)$, which yields more disconnected clusters (3b); the effect is evident when showing the prior and the aggregated posterior that tries to cover it (3c). Figure 3d (leftmost and rightmost columns) shows that every digit $0-9$ is indeed represented on one of the 16 vertices, while some digits are present with multiple styles, e.g. the 7. The central samples in the Figure are the interpolations obtained by sampling on edges connecting vertices – no real data is autoencoded. Samples from the vertices appear much crisper than other prior samples (3e), a sign of mismatch between prior and aggregated posterior on areas with lower probability mass. Finally, we point out that we could even learn the Dirichlet hyperparameter(s) with a reparametrization trick (Figurnov et al., 2018) and let the data inform the model on the best prior.

### 6.5 TOY PROBABILISTIC PROGRAMMING

We run a final experiment to showcase that SAE can handle more complex implicit distributions, on a toy example of probabilistic programming. The goal is to learn a generative model for MNIST digits positioned on a larger canvas; the data is corrupted with salt noise that we do not model explicitly and which are model is thus required to ignore. The generative model samples from a factored prior distribution for $z_{what}$ — the digit appearance — from a 10-dimensional sphere and for $z_{where}$ — the location and scale — from a 3-dimensional Normal. A decoder network is fed with $z_{what}$ and generates the digit; the digit is then positioned on the black canvas on the coordinates given by a spatial transformer (Jaderberg et al., 2015) which is fed with $z_{where}$. The inference model produces $z_{what}, z_{where}$ from the canvas, by using a spatial transformer and a encoder mirroring the generator.

Our autoencoder is fully deterministic. The cost in latent space amounts to the sum of the Sinkhorn distances in the two prior components, Normal and hyperspherical. Figure 4 compares qualitatively with a simplified version of AIR (Eslami et al., 2016), that is built on variational inference with an explicit modelling of the approximate posterior distribution for this program. SAE is able to replicate the behaviour of AIR by locating the digit on the canvas, ignoring the noise in reconstruction and generating realistic samples.

## 7 CONCLUSIONS

We introduced a new generative model built on the principles of Optimal Transport. Working with empirical Wasserstein distances and deterministic networks provides us with a flexible likelihood-free framework for latent variable modeling. Besides, the theory suggests improving matching in latent space which could be achieved by the use of parametric implicit prior distributions.

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

## A  APPENDIX

### A.1  LEMMA

As a useful helper Lemma, we prove a Lipschitz property for the Wasserstein distance $W_p$.

**Lemma A.1.** *For every $P_X, P_Y$ distributions on a sample space $\mathcal{S}$ and a Lipschitz map $F$ we have that*

$$W_p(F_\# P_X, F_\# P_Y) \le \gamma \cdot W_p(P_X, P_Y),$$

*where $\gamma$ is the Lipschitz constant of $F$.*

*Proof.* Recall that

$$W_p(F_\# P_X, F_\# P_Y)^p = \inf_{\Gamma \in \Pi(F_\# P_X, F_\# P_Y)} \int_{\mathcal{S} \times \mathcal{S}} \|x - y\|_p^p \, d\Gamma(x, y).$$

Notice then that for every $\Gamma \in \Pi(P_X, P_Y)$ we have that $(F \times F)_\# \Gamma \in \Pi(F_\# P_X, F_\# P_Y)$. Hence

$$\{(F \times F)_\# \Gamma : \Gamma \in \Pi(P_X, P_Y)\} \subset \Pi(F_\# P_X, F_\# P_Y). \tag{17}$$

From (16) we deduce that

$$\begin{aligned} W_p(F_\# P_X, F_\# P_Y)^p &\le \inf_{\Gamma \in \Pi(P_X, P_Y)} \int_{\mathcal{S} \times \mathcal{S}} \|x - y\|_p^p \, d(F \times F)_\# \Gamma \\ &= \inf_{\Gamma \in \Pi(P_X, P_Y)} \int_{\mathcal{S} \times \mathcal{S}} \|F(x) - F(y)\|_p^p \, d\Gamma \\ &\le \gamma^p \cdot (W_p(P_X, P_Y))^p. \end{aligned}$$

Taking the $p$-root on both sides we conclude. □

## A.2 PROOF OF THEOREM 3.1

*Proof.* Using the triangle inequality of the Wasserstein distance we obtain

$$
\begin{aligned}
W_p(P_X, G_\# P_Z) &\le W_p(P_X, G_\# Q_Z) + W_p(G_\# Q_Z, G_\# P_Z) \\
&\le W_p(P_X, G_\# Q_Z) + \gamma \cdot W_p(Q_Z, P_Z),
\end{aligned} \tag{18}
$$

where in line (17) we have used Lemma A.1. $\qquad\square$

In case $G$ is not deterministic, defining

$$
\gamma = \sup_{\mathcal{P}, \mathcal{Q}} \frac{W_p(G(X|Z)_\# \mathcal{P}, G(X|Z)_\# \mathcal{Q})}{W_p(\mathcal{P}, \mathcal{Q})}
$$

exp we can still formulate a bound. The result follows directly from the first line of (17).

## A.3 PROOF OF THEOREM 3.2

The basic tool to prove Theorem 3.2 is the equivalence between Monge and Kantorovich formulation of optimal transport. For convenience we formulate its statement and we refer to Villani (2008) for a more detailed explanation.

**Theorem A.2** (Monge-Kontorovich equivalence). *Given $P_X$ and $P_Y$ probability distributions on $\mathcal{X}$ such that $P_X$ is not atomic, $c : \mathcal{X} \times \mathcal{X} \to \mathbb{R}$ continuous, we have*

$$
W_c(P_X, P_Y) = \inf_{\substack{T:\mathcal{X}\to\mathcal{X}: \\ T_\# P_X = P_Y}} \int_\mathcal{X} c(x, T(x)) \, dP_X(x). \tag{19}
$$

We are now in position to prove Theorem 3.2. We will prove it for a general continuous cost $c$.

*Proof.* Notice that as the encoder $Q(Z|X)$ is deterministic there exists $Q : \mathcal{X} \to \mathcal{Z}$ such that $Q_Z = Q_\# P_X$ and $Q(Z|X) = \delta_{\{Q(x)=z\}}$. Hence

$$
\begin{aligned}
\mathbb{E}_{X \sim P_X} \mathbb{E}_{Z \sim Q(Z|X)}[c(X, G(Z))] &= \int_{\mathcal{X} \times \mathcal{Z}} c(x, G(z)) \, dP_X(x) d\delta_{\{Q(x)=z\}}(z) \\
&= \int_\mathcal{X} dP_X(x) \int_\mathcal{Z} c(x, G(z)) d\delta_{\{Q(x)=z\}}(z) \\
&= \int_\mathcal{X} c(x, G(Q(x))) \, dP_X(x).
\end{aligned}
$$

Therefore

$$
\inf_{Q(Z|X) \text{ deterministic: } Q_Z = P_Z} \mathbb{E}_{X \sim P_X} \mathbb{E}_{Z \sim Q(Z|X)}[c(X, G(Z))] = \inf_{\substack{Q:\mathcal{X}\to\mathcal{Z} \\ Q_Z = P_Z}} \int_\mathcal{X} c(x, G(Q(x))) \, dP_X. \tag{20}
$$

We now want to prove that

$$
\{G \circ Q : Q_\# P_X = P_Z\} = \{T : \mathcal{X} \to \mathcal{X} : T_\# P_X = P_G\}. \tag{21}
$$

For the first inclusion $\subset$ notice that for every $Q : \mathcal{X} \to \mathcal{Z}$ such that $Q_Z = P_Z$ we have that $G \circ Q : \mathcal{X} \to \mathcal{X}$ and

$$
(G \circ Q)_\# P_X = G_\# Q_\# P_X = G_\# P_Z.
$$

For the other inclusion $\supset$ consider $T : \mathcal{X} \to \mathcal{X}$ such that $T_\# P_X = P_G = G_\# P_Z$. We want first to prove that there exists a set $A \subset \mathcal{X}$ with $P_X(A) = 1$ such that $G : \mathcal{Z} \to T(A)$ is surjective. Indeed if it does not hold there exists $B \subset \mathcal{X}$ with $P_X(B) > 0$ and $G^{-1}(T(B)) = \emptyset$. Hence

$$
0 = G_\# P_Z(T(B)) = T_\# P_X(T(B)) = P_X(B) > 0
$$

that is a contraddiction. Therefore by standard set theory the map $G : \mathcal{Z} \to T(A)$ has a right inverse that we denote by $\widetilde{G}$. Then define $Q = \widetilde{G} \circ T$. Notice that $G \circ Q = G \circ \widetilde{G} \circ T = T$ almost surely in $P_X$ and also

$$(\widetilde{G} \circ T)_{\#} P_X = P_Z \, .$$

Indeed for any $A \subset \mathcal{Z}$ Borel we have

$$(\widetilde{G} \circ T)_{\#} P_X(A) = (\widetilde{G} \circ G)_{\#} P_Z(A) = P_Z(\widetilde{G}^{-1}(G^{-1}(A))) = P_Z(A) \, .$$

This concludes the proof of the claim in (20). Now we have

$$\inf_{\substack{Q:\mathcal{X} \to \mathcal{Z} \\ Q_{\#}(P_X) = P_Z}} \int_{\mathcal{X}} c(x, G(Q(x))) \, dP_X(x) = \inf_{\substack{T:\mathcal{X} \to \mathcal{X} \\ T_{\#}(P_X) = P_G}} \int_{\mathcal{X}} c(x, T(x)) \, dP_X(x) \, .$$

Notice that this is exactly the Monge formulation of optimal transport. Therefore by Theorem A.2 we conclude that

$$\inf_{\substack{Q(Z|X) \text{ deterministic}: \ Q_Z = P_Z}} \mathbb{E}_{X \sim P_X} \mathbb{E}_{Z \sim Q(Z|X)}[c(X, G(Z))] = \inf_{\Gamma \in \Pi(P_X, P_G)} \mathbb{E}_{(X,Y) \sim \Gamma}[c(X, Y)]$$

as we aimed. $\qquad\square$

### A.4 Tolstikhin et al. (2018)'s Theorem as a consequence

*Proof.* Thanks to Theorem 3.2 we have that

$$
\begin{aligned}
W_c(P_X, P_G) &= \inf_{\substack{Q(Z|X) \text{ deterministic}: \ Q_Z = P_Z}} \mathbb{E}_{X \sim P_X} \mathbb{E}_{Z \sim Q(Z|X)}[c(X, G(Z))] \\
&\geq \inf_{\substack{Q(Z|X): \ Q_Z = P_Z}} \mathbb{E}_{X \sim P_X} \mathbb{E}_{Z \sim Q(Z|X)}[c(X, G(Z))] \, .
\end{aligned}
$$

For the opposite inequality given $Q(Z|X)$ such that $P_Z = \int Q(Z|X) dP_X$ define $Q(X, Y) = P_X \times [G_{\#} Q(Z|X)]$. It is a distribution on $\mathcal{X} \times \mathcal{X}$ and it is easy to check that $\pi^1_{\#} Q(X, Y) = P_X$ and $\pi^2_{\#} Q(X, Y) = G_{\#} P_Z$, where $\pi^1$ and $\pi^2$ are the projection on the first and the second component. Therefore

$$\{Q(X, Z) : Q(Z|X) \text{ such that } Q_Z = P_Z\} \subset \Pi(P_X, P_G)$$

and so

$$
\begin{aligned}
W_c(P_X, P_G) &\leq \inf_{Q(Z|X): Q_Z = P_Z} \int_{\mathcal{X} \times \mathcal{X}} c(x, y) \, dQ(x, y) \\
&= \int_{\mathcal{X}} \left[ \int_{\mathcal{X}} c(x, y) \, dG_{\#} Q(Z|X)(y) \right] dP_X \\
&= \int_{\mathcal{X}} \left[ \int_{\mathcal{X}} c(x, G(z)) \, dQ(Z|X)(z) \right] dP_X \, .
\end{aligned}
$$

$\qquad\square$

### A.5 Bounds for neural networks

**Theorem A.3.** *Let $P_X$ be non-atomic and $G(X|Z)$ be deterministic and $\gamma$-Lipschitz. If $Q^{NN}$ is a neural network approximating a near optimal deterministic encoder up to an error of $\varepsilon \geq 0$ in $L_p$-norm then we have the inequality:*

$$0 \leq \left\{ \sqrt[p]{\mathbb{E}_{X \sim P_X}[\|X - G(Q^{NN}(X))\|_p^p]} + \gamma \cdot W_p(Q_Z^{NN}, P_Z) \right\} - W_p(P_X, P_G) \leq 3\gamma\varepsilon.$$

*Proof.* Let $Q^*$ be an optimal measurable deterministic encoder that optimizes the right-hand side of Theorem 3.1 among measurable deterministic encoder (or at least $\delta \leq \gamma\varepsilon$ close to it) and $Q^{NN}$ a

neural network approximation of $Q^*$ such that $\sqrt[p]{\mathbb{E}_{X \sim P_X}[\|Q^*(X) - Q^{NN}(X))\|_p^p]} \leq \varepsilon$ (existence by Hornik (1991)). Then we get:

$$\sqrt[p]{\mathbb{E}_{X \sim P_X}[\|X - G(Q^{NN}(X))\|_p^p]} + \gamma \cdot W_p(Q_Z^{NN}, P_Z)$$

$$\overset{\text{triangle ineq.}}{\leq} \sqrt[p]{\mathbb{E}_{X \sim P_X}[\|X - G(Q^*(X))\|_p^p]}$$

$$+ \underbrace{\sqrt[p]{\mathbb{E}_{X \sim P_X}[\|G(Q^*(X)) - G(Q^{NN}(X))\|_p^p]}}_{\leq \gamma \cdot \varepsilon}$$

$$+ \gamma \cdot \underbrace{W_p(Q_Z^{NN}, Q_Z^*)}_{\leq \varepsilon} + \gamma \cdot W_p(Q_Z^*, P_Z)$$

$$\leq \sqrt[p]{\mathbb{E}_{X \sim P_X}[\|X - G(Q^*(X))\|_p^p]} + \gamma \cdot W_p(Q_Z^*, P_Z) + 2\gamma\varepsilon$$

$$\overset{\text{Def. of } Q^*}{\leq} \inf_Q \left\{ \sqrt[p]{\mathbb{E}_{X \sim P_X}[\|X - G(Q(X))\|_p^p]} + \gamma \cdot W_p(Q_Z, P_Z) \right\} + \delta + 2\gamma\varepsilon$$

$$\overset{3.2}{\leq} W_p(P_X, P_G) + 3\gamma\varepsilon,$$

where in the last inequality we use additionally that the upper bound in Theorem 3.1 is sharp.  □

Finally, we can also formulate a version of Corollary 3.3 restricted to deterministic neural networks as follows:

**Theorem A.4.** *Let $P_X$ be non-atomic and $G(X|Z)$ be deterministic and $\gamma$-Lipschitz. Then we have the equality:*

$$W_p(P_X, P_G) = \inf_{Q \; NN} \sqrt[p]{\mathbb{E}_{X \sim P_X}[\|X - G(Q(X))\|_p^p]} + \gamma \cdot W_p(Q_Z, P_Z),$$

*where $Q$ runs through all deterministic neural network encoders (or any other class of universal approximators).*

*Proof.* This directly follows from A.3 in combination with Hornik (1991).  □

## A.6  PROOF OF THEOREM 3.4

*Proof.* Statement $i$) follows directly from the definition of push-forward of a measure.

For $ii$) notice that if $P_Z \neq Q_Z$ then there exists $A \subset \mathcal{Z}$ a Borel set such that $P_Z(A) \neq Q_{\#}P_X(A)$. Then

$$G_{\#}P_Z(G(A)) = P_Z((G^{-1} \circ G)(A)) = P_Z(A) \neq Q_{\#}P_X(A)$$
$$= Q_{\#}P_X(A)((G^{-1} \circ G)(A)) = (G \circ Q)_{\#}P_X(G(A)).$$

Hence as $P_X = (G \circ Q)_{\#}P_X$ by hypothesis, we immediately deduce that $P_X \neq P_G$.  □

## A.7 COMPARISON WITH BOJANOWSKI & JOULIN (2017)

We prove that the cost function of NAT is equivalent to ours when the encoder output is $L_2$ normalized, $c'$ is squared Euclidean and the Sinkhorn distance is considered with $\varepsilon = 0$:

$$\arg\max_\theta \max_{R \in P_M} \ \mathrm{Tr}(RZf_\theta(X)^\top) \tag{22}$$

$$= \arg\max_\theta \max_{R \in P_M} \ \langle RZ, f_\theta(X) \rangle_F \tag{23}$$

$$= \arg\min_\theta \min_{R \in P_M} \ 2 - 2\langle RZ, f_\theta(X) \rangle_F \tag{24}$$

$$= \arg\min_\theta \min_{R \in P_M} \ \|RZ\|_F^2 + \|f_\theta(X)\|_F^2 - 2\langle RZ, f_\theta(X) \rangle_F \tag{25}$$

$$= \arg\min_\theta \min_{R \in P_M} \ \|RZ - f_\theta(X)\|_F^2 \tag{26}$$

$$= \arg\min_\theta \min_{R \in P_M} \ \sum_{i,j} R_{i,j}\|z_i - f_\theta(x_j)\|_2^2 \tag{27}$$

$$= \arg\min_\theta \min_{R \in P_M} \ \langle R, C \rangle_F \tag{28}$$

$$\subseteq \arg\min_\theta \min_{R \in S_M} \ \tfrac{1}{M}\langle R, C \rangle_F - 0 \cdot H(R) \ . \tag{29}$$

Step 24 holds because both $R$ and $f_\theta(X)$ are row normalized. Step 25 exploits $R$ being a permutation matrix. The inclusion in Step 28 extend to degenerate solutions of the linear program that may not lie on vertices. We have discussed several differences between our Sinkhorn encoder and NAT. There are other minor ones with Bojanowski & Joulin (2017): ImageNet inputs are first converted to grey and passed through Sobel filters and the permutations are updated with the Hungarian only every 3 epochs. Preliminary experiments ruled our any clear gain of those choices in our setting.

## A.8 PROOF OF PROPOSITION 6.1

*Proof.* Let $z, z'$ two points sampled uniformrly from a $d$-dimensional sphere. Let $\alpha$ be the Euclidean distance between the two points. $\alpha$ has an analytical form (Wu et al., 2017) :

$$p(\|z - z'\|_2) = p(\alpha) = \frac{\alpha^{d-2}}{c(d)}\left[1 - \frac{1}{4}\alpha^2\right]^{\frac{d-3}{2}}, \quad \text{where} \quad c(d) = \sqrt{\pi}\frac{\Gamma\left(\frac{d-1}{2}\right)}{\Gamma\left(\frac{d}{2}\right)}.$$

For high dimension, it approaches a Gaussian: $p(\alpha) \approx \mathcal{N}(\sqrt{2}, \frac{1}{2d})$ as $d \to +\infty$. By the Chebischev inequality, for every $t > 0$

$$P(|\alpha - \sqrt{2}| \geq t) \leq \frac{1}{2dt^2} \ .$$

Choosing $t = -\delta + \sqrt{2}$ for $\delta < \sqrt{2}$ and using the symmetry of the Gaussian around the expectation we obtain

$$\frac{1}{2d(\sqrt{2} - \delta)^2} \geq P(|\alpha - \sqrt{2}| \geq -\delta + \sqrt{2})$$

$$= 2P(\alpha \leq \sqrt{2} + \delta - \sqrt{2})$$

$$= 2\left(1 - P(\alpha \geq \delta)\right) \ .$$

Hence

$$P(\alpha \geq \delta) \geq 1 - \frac{1}{4d(\sqrt{2} - \delta)^2} \ .$$

$\square$

