# OpenReview forum: "Sinkhorn AutoEncoders"
_ICLR.cc/2019/Conference_

### Official Review · AnonReviewer2 · 2018-11-05
**Well-written and significant novelty paper**

**Rating:** 7
**Confidence:** 3

**Review:**

The paper proposes a new representation of Wasserstein AutoEncoder and provides the formal analysis of learning autoencoders with optimal transport theory. The proposed model, SAE, employs the constraints on the equality of prior and posterior latent spaces with a Sinkhorn distance. Moreover, the proposed model is also backed up with some theoretical guarantees.

The paper is well-written and easy to follow. The experimental results with different priors have demonstrated the effectiveness of the newly formulated model. However, it is not convinced that what is the advantages of the proposed model with WAE. Can the authors provide more insight and comparison with its counterpart, WAE?

In term of time-complexity, computing Sinkhorn distance in Alg 1 introduces computation overhead, especially with small \epsilon. In compared with WAE, what is the computation overhead of the proposed model? Can the authors provide some theoretical analysis of time-complexity and experimental results?

In Table 2, there are only FID values for WAE with MMD cost. Can the authors show the numbers with WAE-GAN on these datasets?

Conclusion: The theoretical and experimental contributions are significant to publish at the venue.

---

### Official Review · AnonReviewer3 · 2018-11-05
**Contributions should be better explained**

**Rating:** 7
**Confidence:** 3

**Review:**

Summary

This paper builds upon the recent Wasserstein Auto-Encoders; the main innovation is the use of the Sinkhorn distance between the prior and the aggregate posterior. This distance, is used as a differentiable and more tractable surrogate for the wasserstein distance, which is proposed as an alternative to heuristics (e.g. MMD) in Wasserstein-auto encoders.

Also, the authors provide some theoretical results complementing the Wasserstein Auto-Encoders papers and illustrate their results showing better or equal performance than with alternatives, particularly VAE and WAE.

Evaluation.
It is certainly a good paper, and well written (but notice several typos). The experimental part is thorough which certainly plays in favor at evaluation time. Theoretical results are a significant contribution, but not quite deep.

 But what troubles me is that when I read the paper it was hard for me to understand what the contribution is. My guess on this is what I wrote in the Summary. And if that is the case, I think it is a rather marginal contribution: it would be desired to have a theory for why using the wassertein penalization is better than the ones proposed in the WAE paper (Recall, there, penalizations are introduced to get rid of a constraint that appears in the minimization paper). The authors argue that Sinkhorn does as good as wasserstein and I can believe that (but it would be good if authors could refer to recent results [1] regarding the quality of this approximation, which essentially say that sinkhorn does no do miracles) but the most fundamental question is why a wasserstein penalization is better than the ones proposed in WAE. If no such theory is available, it is hard for me to judge the paper as non-marginal.

Theorem 3.1 is an attempt for such an explanation, but it was not clear at all for me why theorem 3.1 implies that wasserstein distance is the penalization to use. Theorem 3.1 only gives an upper bound. The authors may elaborate on this


I hope my criticism is helpful for improving the current version of the manuscript.


[1] http://proceedings.mlr.press/v75/weed18a/weed18a.pdf


=== after rebuttal ===

The authors have addressed my concerns and I have updated my score accordingly.

---

### Official Review · AnonReviewer1 · 2018-11-08
**summary**

**Rating:** 6
**Confidence:** 3

**Review:**

I tried to understand the paper it seems to me that the paper proposed a new objective function for learning an autoencoder based on Sinkhorn algorithm. Following the idea of Sinkhorn algorithm, the Sinkhorn autoencoder minimizes the W-distance between aggregated posterior and the data prior via integrating the objective of (original) autoencoder and Sinkhorn distance. This looks new, but I did not look insight to the detailed derivations. One thing that I do not like is that Sinkhorn distance needs to be optimized before optimizing the autoencoder distance, if I understand correctly.

---

> ### Author Response · Authors · 2018-11-08
> **Reply to AnonReviewer1**
>
> Thank you for your comment. We would like to make some clarity on how we utilize the Sinkhorn algorithm in this work, which is similar to the insight made in [B]. The Sinkhorn composes O(L) operations, which are mostly linear and, more importantly, they are all differentiable. For the purpose of back propagation, we can interpret those as a stack of layers to compute the loss function for the encoder in latent space. We remark that there is no need of alternating optimization — as running the Hungarian algorithm would require. Instead, we differentiate through the Sinkhorn operations to compute an approximation of the Wasserstein distance in latent space.
>
> [B] Genevay, A., Peyré, G., & Cuturi, M. (2017). Learning generative models with sinkhorn divergences. arXiv preprint arXiv:1706.00292.

---

### Official Review · AnonReviewer4 · 2018-11-13
**Good motivation but empirical evidence shows limited improvements.**

**Rating:** 5
**Confidence:** 4

**Review:**

The paper introduces a new cost function for training Wasserstein Autoencoders that combines reconstruction error with Sinkhorn distance on the latent space. Authors provide nice theoretical motivation, yet empirical results seem incremental and do not fully support the effectiveness of this approach.

Pros:
- Theorem 3.1 (although trivial) provides motivation for optimizing Wasserstein distance in the latent space in WAEs.
- Theorem 3.2 shows sufficiency of optimization over deterministic encoders in WAEs.
- The proposed SAE virtually does not favor any prior and can preserve some aspects of geometry of the original space.

Cons:
- It is unclear why Sinkhorn algorithm would provide better estimate of Wasserstein distance than e.g. adversarial WGANGP (which would be a variant of GAN-WAE). Sinkhorn convergence is discussed only in terms of sample size and  smoothing regularizer, not in the context of batch training.
- Quantitative results are on par or marginally better than other methods, they also lack some comparisons (see details below).
- There is no comparison to relevant models outside VAE scope, e.g. ALI [4].

The novelty of this paper is combining WAEs with Sinkhorn algorithm. Overall, it has potential, but the proposed method would probably require clearer evaluation.

Detailed issues:
- Notation for posterior seems somewhat inconsistent and misleading, namely push-forward G#P_Z = P_G, while Q#P_X = Q_Z.
- It is unclear why MMD or GAN losses on WAS's latent space are referred to as heuristics, each of these constitutes a divergence in the same way as the proposed Sinkhorn distance.
- FID scores for MNIST are incomparable due to the use of own network; using LeNet has been proposed [3].
- It is unclear what ‘Empirical lower bounds’ for MMD mentioned in Table 1. caption mean, as unbiased MMD estimator (e.g. [2]) is available. On the other hand, FID is known to be biased [3], so test-set FID should be provided for comparison.
- Table 2. lacks comparison of SAE with normal prior even though a) authors note that MMDs are incomparable with different priors, b) SAEs is claimed to be prior-agnostic, c) in such setting MMD-WAE might be advantageous [1]. Again, no test-set FID scores.
- Samples in Figure 2 too small.
- MMD lacks citation (e.g. [2]).

Typos:
p.6 line 3 construcetion -> construction
p.6 line 30 Hypersherical -> Hyperspherical
P.8 line 1 this a sign -> this is a sign

[1] Ilya Tolstikhin, Olivier Bousquet, Sylvain Gelly, and Bernhard Schölkopf. Wasserstein Auto-Encoders. ICLR 2018.
[2] Arthur Gretton, Karsten M. Borgwardt, Malte J Rasch, Bernhard Schölkopf, and Alex J. Smola. A kernel two-sample test. The Journal of Machine Learning Research, 13, 2012a.
[3] Mikołaj Bińkowski, Dougal J. Sutherland, Michael Arbel, Arthur Gretton. Demystifying MMD GANs. ICLR 2018.
[4] Vincent Dumoulin, Ishmael Belghazi, Ben Poole, Olivier Mastropietro, Alex Lamb, Martin Arjovsky and Aaron Courville. Adversarially Learned Inference. ICLR 2017

---

> ### Author Response · Authors · 2018-11-18
> **We thank the reviewer for the detailed analysis and for the several suggestions for improvement.**
>
> EDIT: more comments on WGAN, heuristics and a correction about the FID scores
> ===
>
> In the updated submission, we are going to show more clearly how Theorem 3.1 is fundamental for the construction of our learning objective. We will also show that the provided bound — which uses a Wasserstein distance as latent space penalty — provides us with a tight bound for learning the generative model. On this regard, a variant of WAE that uses a WGAN as penalty would in fact optimize a similar objective. Although, an interesting point on this regard is raised in [A]: a penalty derived by a GAN would provide a *lower bound*, which is the opposite of what needed in the WAE/SAE minimization problem. Still, it is true that their relative quality could be measured experimentally; yet we are not aware of any published work on a WAE-WGAN and we believe it is unfair to require a novel combination of methods as a comparative baseline.
>
> Additionally, as stated in a review below: we did not compare with a WAE-(W)GAN or ALI as we are interested in deepening the understanding and improving the quality of auto-encoders as in their min-min kind of objective -- this really follows WAE's motivations as well. Notice that we could also extend SAE with an adversarial cost in latent space, by learning the transportation cost c. We believe that only this extension of SAE would be a fair comparison with WAE-GAN. We will leave this direction for future work.
>
> We call WAE-GAN/MMD heuristics relatively to the fact that we can justify the alternative choice of the Wasserstein distance due to Theorem 3.1. We will rephrase this argument. Moreover, when one introduces the WAE's objective by means of Lagrangian multipliers, optimization consists of a minimization of a lower bound of the problem [A].
>
> We did not discuss Sinkhorn's convergence in terms of batch training. Yet previous work such as [B, and others referred in our paper] showed the empirical merits of minibatch training by backpropagating through the Sinkhorn. An analysis would be an interesting addition but we consider it outside the scope of this work.
>
> We will improve the paper presentation in order to make these points clear:
> * Empirical lower bound for MMD: they are computed as the MMD between two (independent) set of N samples from the prior, where N is the same same of the test test. The number gives us a lower bound for the MMD between the aggregated posterior and prior.
> * Missing SAE with normal prior in table 2: while we could have reported those numbers for completeness, we only experimented with hypersphere in the section and still obtained competitive results. We don’t see this as a limitation, as we consciously did not attempt to tune the prior as hyper-parameter in order to improve the results. Please notice that we use several shapes of priors with SAE in other experiments, also Normal (6.5).
> * Thanks for providing a pointer to [3], of which we were not aware. Following the paper perspective, we agree that the FID bias may be relevant in our experimental results. Time permitting (either during this review or for the final version), we will report FID scores on the MNIST test set and study their difference. On the use of a FID-LeNet score, we do not believe that a numerical comparison with other work on the MNIST FID score would be too insightful, due to the simplicity of the dataset. Our strategy and network for computing the FID were inspired by [C].
>
> [A] Bousquet, Olivier, et al. "From optimal transport to generative modeling: the VEGAN cookbook." arXiv preprint arXiv:1705.07642 (2017).
> [B] Genevay, A., Peyré, G., & Cuturi, M. (2017). Learning generative models with sinkhorn divergences. arXiv preprint arXiv:1706.00292.
> [C] Odena, Augustus, et al. "Is Generator Conditioning Causally Related to GAN Performance?." arXiv preprint arXiv:1802.08768 (2018).

---

> > ### Comment · AnonReviewer4 · 2018-12-10
> > **Some clarifications are valid, but empirical evidence is still missing**
> >
> > Authors have addressed some of my concerns, yet in most cases the response is limited to defending their positions. It is clear now for me that the paper provides deeper insights into WAE and proposes promising alternative of SAE.
> >
> > However, the empirical results are still missing and there are some major flaws:
> > 1) Missing normal prior in Table 2. I see this as a limitation; it is the authors who claim that SAE is prior-agnostic, so results with different priors are expected. Choosing one prior actually looks like hyperparameter cherry-picking.
> > 2) Some model variants are evaluated (e.g. Sinkhorn vs Hungarian), but comparisons against competitive models or with varying network architectures are missing.
> > 3) Scores are ambiguous. Lower bound on MMD between two samples from the same prior does not exist (Theorem 10, [2] above). Also, if FID comparison on MNIST is not too insightful, why are these experiments reported?
> >
> > I believe that papers which build upon a successful idea such as WAE need to show empirical evidence that the proposed new theory leads to improved results (a good example would be presentation of superiority of WGAN-GP [Gulrajani et al.] over WGAN [Arjovsky et al.]). Although I like the idea of SAE, I believe this paper requires further experiments and therefore is slightly below the acceptance threshold.

---

### Author Response · Authors · 2018-11-07
**Reply to AnonReviewer2 and AnonReviewer3**

We thank the reviewers for the analysis of our work.

We see our theory, and in particular the statement in Theorem 3.1, as a fundamental contribution of our work, which goes beyond a simple extension of the WAE framework. The issues raised by the reviewers give us the opportunity for a more detailed comment in support of the use of the Wasserstein distance in latent space. We will work on a better presentation of the following arguments for the camera ready.

Recall that WAE uses either a MMD or implements a discriminator which minimizes a JS-divergence, i.e symmetrized KL divergence. Instead, the use of a Wasserstein distance is relevant because:
- Wasserstein is upper bounded by JS. See Theorem 2 in [A] . This property is often mentioned as Wasserstein inducing a weaker topology with respect to JS. While small JS implies small Wasserstein, the converse is not true and therefore minimizing Wasserstein can be beneficial in practice.
- Sinkhorn takes the best of Wasserstein and MMD, and indeed can be characterized as interpolating between the two. See [B] and the more recent [C].

The use the Wasserstein distance in the latent space of a generative auto-encoder is theoretically justified by Theorem 3.1. The validity of Theorem 3.1 relies on the properties of the Wasserstein distance. In fact, the triangle inequality is only possible for metrics, not for general divergencies like KL.

Moreover the bound in Theorem 3.1 is sharp: if one of the terms is zero then the inequality is actually equality. Furthermore, the Lipschitz constant inequality is also tight: if aggregated posterior and prior match, the actual term of the triangle inequality is zero and again equality follows — see also Theorem 3.3.

The arguments are abstract, but simple, and directly give a consistent optimization objective in one framework. In contrast, WAE makes and elegant and detailed analysis first, but finally resorts to Lagrange multipliers with ad-hoc divergences, therefore partially abandoning the framework of Optimal Transport.

We thank AnonReviewer3 for pointing out the analysis of [1]. We referred to [D] which also provide finite sample guarantees. We will update our references.

Regarding the specific questions of AnonReviewer2, we did not compare with a WAE-GAN as we are interested in deepening the understanding and improving the quality of auto-encoders as in their min-min kind of objective -- this really follows WAE's motivations as well. Notice that we could also extend SAE with an adversarial cost in latent space, by learning the transportation cost c. We believe that only this extension of SAE would be a fair comparison with WAE-GAN. We will leave this direction for future work.

The computational overhead of running the Sinkhorn with respect to, e.g. MMD, is always relative to the neural network architecture. With CelebA and CIFAR10 we train high capacity networks and the cost of running the Sinkhorn is negligible. One can interpret its computation as stacking several (mostly linear) layers on top of the encoder, with a width that is proportional to the batch size. Experiments with very small \epsilon, which can be expensive due to a large L, did not show a clear positive impact on generative performance of our models; thus we did not consider running the Sinkhorn with small entropy regularization. In summary, we did not observe large overhead, except when the neural network is very simple, such as in the toy experiments. We will include an empirical analysis in final version.


[A] Arjovsky, M., Chintala, S., & Bottou, L. (2017). Wasserstein gan. arXiv preprint arXiv:1701.07875.
[B] Genevay, A., Peyré, G., & Cuturi, M. (2017). Learning generative models with sinkhorn divergences. arXiv preprint arXiv:1706.00292.
[C] Feydy, J., Séjourné, T., Vialard, F. X., Amari, S. I., Trouvé, A., & Peyré, G. (2018). Interpolating between Optimal Transport and MMD using Sinkhorn Divergences. arXiv preprint arXiv:1810.08278.
[D] Weed, J., & Bach, F. (2017). Sharp asymptotic and finite-sample rates of convergence of empirical measures in Wasserstein distance. arXiv preprint arXiv:1707.00087.

---

### Public Comment · ~Ilya_Tolstikhin1 · 2018-11-13
**Interesting theoretical insights on the auto-encoder based generative modeling, but questions regarding empirical results.**

I enjoyed reading this paper and I think it provides new theoretical insights on the generative modeling from the optimal transport point of view, and in particular on the WAE model introduced in [1]. The authors propose to use WAE with the Sinkhorn penalty in the latent space, resulting in a new variant of WAE that the authors call SAE.

One important contribution is Theorem 3.1, where the authors theoretically support one particular choice of the divergence term in the WAE objective. It shows that if one uses a p-Wasserstein distance as the latent space penalty in the WAE algorithm, then the corresponding WAE objective is an upper bound on the original p-Wasserstein distance between the data and the model distributions. The proposed SAE algorithm does exactly that by approximating the p-Wasserstein divergence in the latent space with the Sinkhorn algorithm.

Theorem 3.2 is also interesting: it is largely based on a somewhat surprising fact that, in particular, it is possible to find a *measurable* function Q such that Q*N(m) = N(n) with n >> m, where N(d) is a d-dimensional standard normal distribution and g*P is a push-forward of distribution P through the function g. This equation is of course not possible if Q is constrained to be continuous (because, roughly speaking, you can not precisely fill the 2 dimensional space with a 1-dimensional curve) and it was very interesting to learn that this equation can be achieved at all. However, I would like to point out that in a context of the generative modeling, where the encoder function Q is typically implemented as a deep neural network and thus *is continuous*, the feasible set of constrained optimization in Eq. 8 is *empty* (unless the latent space dimensionality precisely matches the intrinsic data dimensionality, which is very strong assumption to put).

However, I have an important question regarding the empirical performance of the proposed SAE method. I would like to address Table 2, which is a summary of the main experiments comparing the proposed SAE with other existing algorithms. In particular, for the CelebA dataset the authors report FID=61 score of WAE-MMD, while the same method (WAE-MMD) on the same dataset with exactly the same architecture (as the authors say on Page 6) was reported to achieve FID=55 in [1] (with a source code available). The proposed SAE achieves 56 on CelebA, which shows that SAE and WAE actually perform on par, at least on CelebA (which is unfortunately the only dataset for which [1] evaluated the FID scores). And in a way it is not too surprising because the objectives of WAE-MMD and SAE are *identical* up to a different choice of the penalty terms.


In summary, it is not clear whether SAE (aka WAE-Sinkhorn) provides improvements compared to the other WAE variants (WAE-MMD or WAE-GAN). The statement that SAE / WAE-Sinkhorn “consistently outperforms” other methods is too strong and not enough supported by the current empirical results.

[1] Tolstikhin et al., Wasserstein Auto-Encoders, ICLR 2018.

---

> ### Author Response · Authors · 2018-11-18
> **On fitting distributions with continuous functions and novelty**
>
> Thank you for the insightful comments.
>
> We agree that in general it would be very difficult to fit a distribution in high dimensions with a distribution coming from a low dimensional space using (continuous) neural networks. But note that the encoder maps from the high dimensional image space to the low dimensional latent space, which is a much easier task. So the feasible set of the constraints are not necessarily empty and much easier to approximate using neural networks. For example even though finding a neural network that fits a 2-dimensional Gaussian from 1-dimensional one is only possible in the high capacity limit of a neural network, fitting a 1-dimensional Gaussian from a 2-dimensional Gaussian only needs a projection and a linear map to match.
>
> In response to the comment that "WAE-MMD and SAE are *identical* up to" the penalty terms: in the stated generality all constructions based on variational auto-encoders are identical up to the penalty term: reconstruction plus some form of matching (aggregated) posterior and prior plus other divergence quantities. In this regard, any WAE variant could be seen as implementation of the idea behind adversarial auto-encoders [A], up to the variations of the penalty term. Yet, WAE is built on an elegant and insightful derivation justified by the use of Wasserstein distances. We believe our theoretical derivation provides an even stronger motivation for the construction of a learning objective as we formulate it.  Furthermore, using the Wasserstein penalty allows us to give a theoretical bound on the error. In the updated submission we will analyse more in detail this bound, showing its tightness, and behaviour in the case of deterministic neural network encoders as model class as well.
>
> [A] Makhzani, Alireza, et al. "Adversarial autoencoders." arXiv preprint arXiv:1511.05644 (2015).

---

### Author Response · Authors · 2018-11-21
**Updated paper**

We have uploaded a revised version of our submission and we kindly invite the reviewers to take it into consideration. Following the reviewers' feedback, we have improved Section 3 and clarified how one obtains our objective function in a principled manner. In particular, one can rewrite W(Px, Pg) by *an equality* with the sum of (p-th root of) reconstruction error and W(Qz, Pz). This is now Corollary 3.3, which follows from the two main Theorems. No such equality for a penalized reconstruction is known for WAE, and even more [A] argues that any soft constraint to enforce Qz =~ Pz yields a minimization of a lower bound.

[A] Bousquet, Olivier, et al. "From optimal transport to generative modeling: the VEGAN cookbook." arXiv preprint arXiv:1705.07642 (2017).

---

### Meta-Review · Area_Chair1 · 2018-12-18
**Revise and resubmit**

**Confidence:** 4
**Recommendation:** Reject

**Metareview:**

The reviewers appreciated the contribution of combining Wasserstein Autoencoders with the Sinkhorn algorithm.

Yet R4 as well as the author of the WAE paper (Ilya Tolstikhin) both expressed concerns about the empirical evaluation.

While R1-R3 were all somewhat positive in their recommendation after the rebuttal, they all have somewhat lower confidence reviews, as is also clear by their comments.

The AC decided to follow the recommendation of R4 as they were the most expert reviewer. The AC thus recommends to "revise and resubmit" the paper.